# Epigenetic Regulation of Nitrogen Signaling and Adaptation in Plants

**DOI:** 10.3390/plants12142725

**Published:** 2023-07-21

**Authors:** Hao Zhang, Xiaoyu Zhang, Jun Xiao

**Affiliations:** 1State Key Laboratory of Plant Cell and Chromosome Engineering, Institute of Genetics and Developmental Biology, Chinese Academy of Sciences, Beijing 100101, China; haozhang@genetics.ac.cn (H.Z.); zhangxiaoyu@genetics.ac.cn (X.Z.); 2University of Chinese Academy of Sciences, Beijing 100049, China; 3Hebei Collaboration Innovation Center for Cell Signaling, Shijiazhuang 050024, China; 4Centre of Excellence for Plant and Microbial Science (CEPAMS), JIC-CAS, Beijing 100101, China

**Keywords:** nitrogen signaling, low-nitrogen adaptation, epigenetic regulation, NUE

## Abstract

Nitrogen (N) is a crucial nutrient that plays a significant role in enhancing crop yield. Its availability, including both supply and deficiency, serves as a crucial signal for plant development. However, excessive N use in agriculture leads to environmental and economic issues. Enhancing nitrogen use efficiency (NUE) is, therefore, essential to minimize negative impacts. Prior studies have investigated the genetic factors involved in N responses and the process of low-nitrogen (LN) adaptation. In this review, we discuss recent advances in understanding how epigenetic modifications, including DNA methylation, histone modification, and small RNA, participate in the regulation of N response and LN adaptation. We highlight the importance of decoding the epigenome at various levels to accelerate the functional study of how plants respond to N availability. Understanding the epigenetic control of N signaling and adaptation can lead to new strategies to improve NUE and enhance crop productivity sustainably.

## 1. Introduction

N is a primary macronutrient for plant growth, and it is a vital component of DNA, RNA, and proteins. Conventionally, farmers have used synthetic fertilizers to provide sufficient N levels to the crops [1]. However, the excessive use of N fertilizers led to negative effects on the environment, such as soil degradation, pollution, and air quality reduction, among other concerns [2,3]. Moreover, over-reliance on N fertilizers resulted in economic issues for farmers. Thus, improving the NUE in crop plants is an urgent challenge for modern agriculture, along with reducing the environmental impact of excessive N use.

The uptake and assimilation of N are essential processes that ensure optimal nutrient acquisition, storage, and metabolism in plants [4,5]. These processes are tightly regulated to guarantee that the plants can fine-tune their N metabolism to alter their response to fluctuations in N availability in the environment. Significant strides have been made in understanding the molecular mechanisms that underpin the transcriptional regulation of N uptake [6], and assimilation [7,8]. Several studies have identified N-sensing pathways and intricate signaling pathways involved in the control of N metabolism [9,10]. Additionally, transcription factors (TFs) that regulate the expression of genes required for N metabolism have also been identified, such as *PCF (TCP)-DOMAIN FAMILY PROTEIN 20 (TCP20)* [11] and *TGACG MOTIF-BINDING FACTOR 1 (TGA1)* [12]. The interplay between the different signaling pathways and TFs involved in the regulation of N metabolism creates a complex network of gene regulation [13]. This guarantees that the plant’s response to N availability is fine-tuned, intricate, and adapts to its environment.

Transcriptional regulation in plants is not only controlled by genetic factors, but also in an epigenetic manner. Epigenetics refers to the process where chemical or structural modifications alter the chromatin state, impacting gene expression without altering the underlying DNA sequence. DNA methylation, histone modifications, enhancer-promoter 3D structural interactions, and noncoding RNA-mediated regulation are examples of epigenetic mechanisms that can affect gene expression [14,15]. In recent years, several studies have shown the critical role of epigenetic regulation in controlling the plant response to environmental cues, including nutrient availability [15,16]. Chromatin remodeling and inter/intrachromosomal DNA-DNA interaction can lead to changes in gene expression, making epigenetics an essential mechanism for plants to adapt to changes in their environment [17,18,19]. Studies on the epigenetic regulation of N signaling and adaptation have shown that N availability can influence DNA methylation, histone modifications and the expression of non-coding RNAs (ncRNAs) in plants [20,21,22,23]. Therefore, understanding the role of epigenetic regulation in response to N availability can provide insights into the molecular mechanisms underlying N metabolism and how plants adapt to changes in N availability.

In this review, we summarize current knowledge of epigenetic regulation under different N conditions and propose decoding the epigenome at various levels provides an opportunity to accelerate the functional study of how plants respond to N availability, and thus improving NUE in crops.

## 2. Epigenetic Regulation in N Responses

Plants possess the remarkable ability to promptly perceive nutrient availability, initiating a cascade of responses to maintain nutrient homeostasis [24]. N, a vital nutrient, is sensed by a transceptor, such as NITRATE TRANSPORTER 1.1, (NRT1.1) [25,26] or NIN LIKE PROTEIN 7 (NLP7) [27]. Through the involvement of signal “messengers”, such as calcium and TFs, N triggers rapid changes in plant transcriptional responses, often occurring as quicky as 3 min after nitrate exposure [28], These responses encompass diverse processes, including nitrate uptake, root growth, and plant architecture [9,10,29]. Recent studies have shown that epigenetic regulation is also involved in these processes.

### 2.1. Epigenetic Regulation of N Uptake

N metabolism in plants encompasses processes such as uptake, transportation, and assimilation, all of which are subject to precise and dynamic regulation by NMGs [9,10]. Notably, the expression of N transporters, such as NRT2.1, can be rapidly induced within just one hour in *Arabidopsis* [30]. Altering the expression levels of genes related to N transport, assimilation, and signaling using transgenic approaches can significantly enhance crop yield or NUE [31,32,33,34,35]. Recent studies have highlighted the impact of chromatin regulation, including histone modification and chromatin structure, in modulating the activity of N transporters [17,23,36].

In dicots, histone modification, such as tri-methylation of lysine 36 or lysine 27 on histone H3 (H3K36me3, H3K27me3) are implicated in the regulation of N uptake process. In *Arabidopsis*, the H3K36me3 ‘writer’ SET DOMAIN GROUP8 (SDG8) plays a significant role in regulating genes related to energy metabolism, including NMGs and photosynthesis, through H3K36me3 modification [21]. *SDG8* is induced by N treatment in roots, particularly in the lateral root cap and pericycle [37]. Recent studies have shown that *SDG8* mediates N-triggered gene-expression reprogramming, including *NRT2.1*, *NRT1.5*, and *GLUTAMINE SYNTHETASE 1;4* (*GLN1;4*) [38] (Figure 1). *sdg8* mutants have shown lower nitrate acquisition than wild-type plants under high N conditions [38]. In tomatoes (*Solanum lycopersicum*), the homologs of *SDG8*, namely *SlSDG33* and *SlSDG34*, regulate *SlNRT1.1* in an N-dependent manner that is associated with the root response to N treatments [36] (Figure 1). Moreover, polycomb repressive complex 2 (PRC2), responsible for the addition of methyl groups to H3K27, directly targets and downregulates *AtNRT2.1* expression in *Arabidopsis* [6]. In high N conditions, HIGH NITROGEN INSENSITIVE 9/ interacts with SUPT6H, and CTD Assembly Factor 1 (HNI9/AtIWS1) recruits PRC2 to the *AtNRT2.1* locus to repress its transcription [23] (Figure 1). Moreover, *AtHNI9* control reactive oxygen species (ROS) homeostasis under conditions of excess N. The mutation of *AtHNI9* leads to elevated ROS levels and ROS-dependent upregulation of *AtNRT2.1*, indicating the role of *HNI9* in balancing N uptake and ROS levels in response to N supply [39].

In monocots, N uptake was regulated by changes in chromatin rearrangement. *CHROMATIN REMODELING COMPLEX SUBUNIT B 101* (*CHB101*) encodes a subunit of the ATP-dependent chromatin remodeling complex known as the SWITCH/SUCROSE NONFERMENTING (SWI/SNF) complex [40,41]. In maize, *ZmCHB101* has been found to govern the expression of nitrate transport genes, including *ZmNRT2.1* and *ZmNRT2.2*, in response to nitrate supply [17] (Figure 1). The *ZmCHB101-RNAi* lines showed significantly reduced nucleosome densities at the −1 and +1 nucleosome regions of *ZmNRT2.1* and *ZmNRT2.2* as compared to WT under nitrate treatment. These low nucleosome densities resulted in enhanced ZmNLP3.1 binding to the promoter regions of *ZmNRT2.1* and *ZmNRT2.2*, further regulating their expression [17] (Figure 1).

### 2.2. Epigenetic Regulation of Root Architecture in Response to N

The N signal greatly influences the root system, which serves as the sensory organ for detecting external nitrogen. Nitrate, in particular, has been shown to modulate various aspects of root development, including primary root growth [42,43], lateral root initiation and elongation [44,45,46,47].

Recent discoveries highlight the regulatory role of microRNAs (miRNAs) on key TFs involved in root development in monocots. One such TF is ARABIDOPSIS NITRATE REGULATED 1 (ANR1), a member of the MIKC-type MADS (MCM1/AGAMOUS/DEFICIENS/SRFl) box family of TFs, which mediates the root response to external N [48]. It stimulates lateral root growth, leading to an increase in lateral root number and length, as well as total shoot fresh weight [49]. The conservation of ANR1 function has been observed in various species, including rice [50] and chrysanthemum [51]. Overexpression of miR444a in rice reduced lateral root elongation in a nitrate-dependent manner by targeting *ANR1* [52] (Figure 1). However, miR444a overexpression promoted primary and adventitious root growth and improved nitrate accumulation under high nitrate concentration conditions [52].

In dicots, in particular *Arabidopsis*, the presence of several ncRNAs has been observed in response to N. The miR393 also regulates primary root length and lateral root density in response to N availability in *Arabidopsis* by targeting *AUXIN SIGNALING F-BOX 3* (*AFB3*) [53]. Similarly, miR167 inhibits lateral root outgrowth in response to N by targeting *AUXIN RESPONSE FACTOR 8* (*ARF8*) in pericycle cells in *Arabidopsis* [37] (Figure 1). Additionally, *T5120*, a long non-coding RNA modulated by NLP7 and NRT1.1, promotes nitrate assimilation and root growth (primary root length and lateral root number) in *Arabidopsis* [22] (Figure 1). Interestingly, the miRNA/target modules are responsive to downstream N metabolites, with miR393 and *ARF8* being induced by an N metabolite produced during N reduction, coordinating plant growth and development in response to both external and internal N availability [53].

### 2.3. Epigenetic Regulation of Plant Architecture in Response to N

N supply exerts an epigenetic influence not only on root development but also on plant architecture, particularly for tillering in monocots. In rice, the application of N fertilizer induces changes in the genome-wide H3K27me3 pattern through NITROGEN-MEDIATED TILLER GROWTH RESPONSE 5 (NGR5). This mechanism facilitates the recruitment of PRC2, thus promoting the repression of branching-inhibitory genes (*Dwarf14* (*D14*) and *SQUAMOSA-promoter binding protein-like 14* (*SPL14*)) through H3K27me3 modification [54] (Figure 1). Interestingly, PRC2 has been reported to associate with the regulation of rice tillering with normal N condition. In particular, VIN3-LIKE 2 (OsVIL2) suppresses the expression of *TEOSINTE BRANCHED1* (*OsTB1*), which is a branching-inhibitory gene, by recruiting PRC2 and promoting bud outgrowth [55]. In this context, NRG5 acts as a missing link, connecting N signaling with the regulation of tillering through PRC2. Consequently, exploring the connection between N signaling and chromatin response to N can uncover crucial factors, like NGR5, involved in the regulation of NUE.

In conclusion, the integration of epigenetic regulations involving histone modification, chromatin remodeling, and miRNA regulation in a plant’s response to N underscores the profound influence of N on the epigenome, transcriptional processes, and subsequent divergence in growth and metabolism. Moreover, establishing the link between N signaling and chromatin response holds great significance. Considering the findings discussed above, exploring the role of epigenomic changes in plants adapted to different N conditions, such as LN, becomes particularly intriguing.

## 3. Epigenetic Regulation Facilitates LN Adaptation

Under LN conditions, plants undergo significant transcriptional reprogramming and developmental alteration [56,57]. The reshaping of plant architecture emerges as a common adaptive strategy, representing a long-term response to LN environments. That includes changes in plant height [58], tiller number [58], and root architecture [59]. Currently, the exploration of epigenetic regulation in LN adaptations predominantly focuses on root architecture regulation, with limited studies examining other processes. Modifying root architecture, particularly under conditions of limited nitrogen availability, becomes vital for efficient nutrient uptake and offers a promising avenue to improve nitrogen-use efficiency [60,61,62,63].

### 3.1. Histone Modification Reshapes Root Architecture for LN Adaptation

In dicots, N condition plays a role in the regulation of root system architecture (RSA) through interactions with auxin transport [59,64]. Notably, the process of auxin transport itself is subject to regulation by histone modification. Histone deacetylase inhibitors (HDIs), which induce the accumulation of hyperacetylated nucleosome core histones in chromatin, have been discovered to regulate the degradation of PIN-FORMED 1 (PIN1). But, the link between the degradation of PIN1 and histone change remains unclear, which may include other players regulated by histone acetylation. This, in turn, inhibits primary root elongation and lateral root emergence in *Arabidopsis* [65]. Furthermore, the deposition of the repressive mark H3K27me3 by PRC2 at the PIN1 locus obstructs founder cell establishment during lateral root initiation [66] (Figure 2).

In monocots, recent research has shed light on the divergent strategies employed by different wheat varieties under LN conditions through histone modifications [67]. In the case of Kenong9204 (KN9204), the gain of H3K27ac (acetylation of lysine 27 on histone H3), such as in *TaPILS7_6D* (PIN-LIKES 7, an auxin efflux carrier), coupled with a reduction in H3K27me3 modifications, work in synergy to enhance the expression of genes associated with root development under LN conditions (Figure 2). Conversely, in Jing411 (J411), nitrate uptake transporters were activated through the gain of H3K27ac and loss of H3K27me3 modifications [67] (Figure 2). Furthermore, altering the H3K27me3 modification pattern through the knockout of SWINGER (SWN), a component of the PRC2 complex, alters the strategy for root development and N uptake in response to LN constraints [67].

### 3.2. DNA Methylation Regulates Root Architecture and Trans-Generation LN Stress Memory

Regulation of auxin biosynthesis in the roots involves DNA methylation. Under LN conditions, the biosynthesis of auxin in the roots of *Arabidopsis* seedlings [68] and wheat [69] is induced. In *Arabidopsis*, LN-induced lateral root growth has been shown to be dependent on *TRYPTOPHAN AMINOTRANSFERASE RELATED 2* (*TAR2*), the auxin biosynthesis gene [68]. Moreover, *TAR2* has also been reported to be a target of RNA-directed DNA methylation (RdDM) in *Arabidopsis* [70] (Figure 2).

DNA methylation have been discovered to play a crucial role in trans-generational stress memory, allowing plants (especially monocots/crops) to inherit adaptive traits to cope with N-deficiency stress. In rice, for instance, modified cytosine methylation patterns have been identified as the key mediators enabling the stable transmission of adaptive traits under N-deficiency stress to the progeny [20]. The offspring of N-deficient-stressed rice plants exhibit enhanced tolerance to N-deficiency-induced stress, evident from their increased plant height, greater whole-plant dry weight, and elevated total N content [20]. Apart from DNA methylation, there are currently no reports regarding other forms of epigenetic regulation involved in the transgeneration “memory” of the N-deficiency stress. A recent study employed the CRISPR-dCas9-TET1 system to selectively remove methylated CG (mCG) and CHG (mCHG) marks, which eventually enhance adaptation to the environment as these marks are involved in the regulation of defense-related genes [71]. By manipulating these heritable epigenetic modifications, it becomes possible to enhance stress tolerance in offspring, presenting a novel avenue for bolstering crops’ resilience to N-deficiency stress.

### 3.3. miRNA Mediated Regulation of Root Growth under LN Conditions

For adaptation to LN environment, recent studies have shed light on the intricate involvement of miRNAs in the regulation of root architecture in *Arabidopsis*. The differential expression of specific miRNAs, such as miR169, miR171, miR395, miR397, miR398, miR399, miR408, miR827, and miR857, has been observed in *Arabidopsis* in response to N starvation [72]. Notably, miR160, miR167, and miR171 have emerged as key regulators of root system growth in *Arabidopsis* in response to N starvation [72]. In wheat, a key TF called *NUCLEAR FACTOR Y A-B1* (*NFYA-B1*) has been identified as playing a pivotal role in promoting root development and enhancing the expression of nitrate transporters in response to LN conditions [73]. Interestingly, miRNA169 has been found to regulate the expression of *TaNFYA-B1* by cleaving its mRNA [73] (Figure 2). This mechanism provides a novel insight into how miRNAs influence root development and nutrient uptake of crops by modulating the expression of critical TFs in response to changes in N availability.

In summary, various forms of epigenetic regulation exert influence over different biological processes, exemplified by the impact of histone modification on the auxin transport process, the involvement of DNA methylation in auxin biosynthesis and trans-generation memory, and the targeting of key TFs by miRNAs. These regulatory mechanisms collectively aid plants in adapting to LN conditions.

## 4. Future Perspectives

In summary, plants utilize various epigenetic mechanisms, such as histone modifications, non-coding RNAs (ncRNAs), and DNA methylations, to orchestrate morphological developmental changes and adapt their N metabolism in response to environmental N availability. Recent evidence emphasizes the functional significance of PRC2-TF regulatory modules as potential targets for enhancing NUE in crops. Nevertheless, several questions remain unanswered. Firstly, it is crucial to elucidate how N signaling triggers chromatin-level responses. Additionally, understanding the specific “mediators,” including *trans* factors and *cis*-elements, that drive dynamic changes in the chromatin landscape and regulate different aspects of plant development is essential. Moreover, determining the contribution of epigenetic regulation to natural variations in NUE among different wheat varieties is vital. Finally, exploring how we can leverage epigenetic regulation to improve NUE and ultimately increase wheat yield while reducing the need for excessive nitrogen fertilization represents an important research objective (Figure 3).

### 4.1. From N Signaling to Chromatin Response

Both plants and animals heavily rely on nutrients and energy for their developmental processes, with nutrient signaling directly impacting the epigenome [15,74]. A recent study has uncovered the glucose-TOR-FIE-PRC2 signaling pathway, in which direct phosphorylation by TOR promotes the translocation of FIE into the nucleus and regulates stem cell genes [75], emphasizing the role of nutrient signaling, particularly carbon resources, in orchestrating plant development through direct chromatin modifications. However, a direct molecular connection between the N signaling network and chromatin regulation has yet to be established (Figure 3). Upon N stimulation, an intricate transcriptional regulatory network is activated, involving transporter-receptors and multiple transcription factors that perceive and transmit environmental N signals [76]. Exploring the direct interactions between N signaling cascade factors and chromatin modifiers, such as writers, erasers, readers of histone modifications, or chromatin remodelers, holds promise as a potential research direction. Such interactions may involve transcriptional regulation, protein–protein interactions, post-translational modifications, and other regulatory mechanisms.

### 4.2. Uncovering the ‘Mediators’ for Specific Chromatin Dynamic in Response to N

Rather than inducing global alterations in chromatin modification, the availability of N tends to affect specific genomic regions and subsets of genes, playing a crucial role in shaping the developmental programs of particular tissues [23,67]. This influence is likely mediated by trans factors, such as NGR5 in rice, which preferentially regulate H3K27me3 dynamics to control tillering [54]. To systematically investigate this phenomenon, a strategy can be employed to identify the genomic regions responsible for N responsiveness. By uncovering conserved sequences and *cis*-elements within these regions, the *trans*-factors involved can be identified through DNA-protein interaction assays. Additionally, protein interaction assays, as demonstrated in Arabidopsis [77], can help establish connections between trans-factors and individual chromatin modifiers, including writers, erasers, and readers.

Furthermore, the response of different root parts to N availability exhibits cellular specificity, suggesting unique functions for each cell type [37]. Single-cell transcriptome data have revealed cell-type-specific nitrate-responsive genes in maize root tips [78], emphasizing the importance of studying cell-type-specific chromatin dynamics in response to N signals through single-cell omic approaches. Such investigations are likely to increase our chances of identifying specific mediators involved in chromatin dynamics in response to N.

### 4.3. Exploring the Epigenetic Regulation in Contributing to Natural NUE Diversity

The use of GWAS analysis is widespread in identifying factors that regulate important agronomic traits and mining elite allelic variations in key regulators [79]. However, this approach primarily focuses on DNA variations within coding regions or proximal promoter regions. It is essential to recognize that variations in DNA methylation status, rather than the sequence itself, can also be associated with traits in crop plants [80], such as domestication in rice [81]. For instance, a naturally occurring DNA methylation variation in the promoter of Colorless non-ripening (Cnr) locus is associated with the fruit ripening in tomato [82], demonstrating the significance of DNA methylation in trait association. By analyzing population-wide DNA methylome information across different crop varieties, we can identify differentially methylated DNA sequences associated with crop development in response to N, which is referred to as epi-GWAS.

In maize, a distal enhancer located approximately 63.9 kb upstream of the transcription start site (TSS) regulates the expression of teosinte branched1 (TB1), influencing shoot branching [83]. Moreover, epigenomic profiles have revealed the presence of distal regulatory elements that exhibit specific and dynamic activities, fine-tuning gene expression both developmentally [84] and in response to N [67]. In addition, three-dimensional genomic interaction techniques can provide insights into the functional roles of distal regulatory regions by establishing connections with their corresponding gene targets [85]. Consequently, profiling the epigenetic landscapes of diverse crops with varying NUE can facilitate the identification of key regulatory regions associated with N utilization, a concept known as reg-GWAS (Figure 3).

### 4.4. Application of Epigenetic Regulation in Improving NUE of Crops

Understanding the molecular connections between N signaling and chromatin response can enhance our knowledge of developmental regulation. By targeting specific chromatin modifiers and factors involved in the response to N availability, we can precisely control gene expression, ultimately increasing crop yields. Exploring the differential DNA methylation region, identification of distal regulatory regions and their interactions with genes through population genetics, and 3D genomics provides valuable insights for optimizing N utilization in crops. With these advancements, CRISPR-based genetic and epigenetic editing that targets genes, regulatory regions, or epigenetic marks have the potential to revolutionize crop improvement, promoting sustainable agriculture practices and minimizing environmental impacts.

## Figures and Tables

**Figure 1 plants-12-02725-f001:**
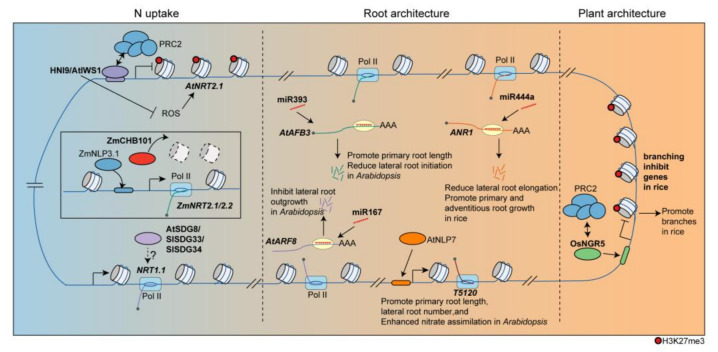
Epigenetic regulation of divergent processes in response to N. HNI9/AtIWS1 binds to the *AtNRT2.1* locus and recruits PRC2 to suppress its expression. Conversely, ZmCHB101 regulates *ZmNRT2.1* by reducing nucleosome densities. The homologs of *SDG8* in tomatoes, *SlSDG33* and *SlSDG34*, also regulate *NRT1.1* under N supply. Epigenetic factors are involved in the regulation of N-dependent development processes as well. And miR444a reduces lateral root elongation by targeting *ANR1* in rice. In *Arabidopsis*, miRNAs, such as miR167 and miR393, target transcripts involved in root development, particularly in response to N, thereby influencing root growth. Additionally, NLP7 modulates the expression of *T5120*, a long non-coding RNA. In rice, OsNGR5, which recruits PRC2 to inhibit branching-inhibition genes, influence plant architecture.

**Figure 2 plants-12-02725-f002:**
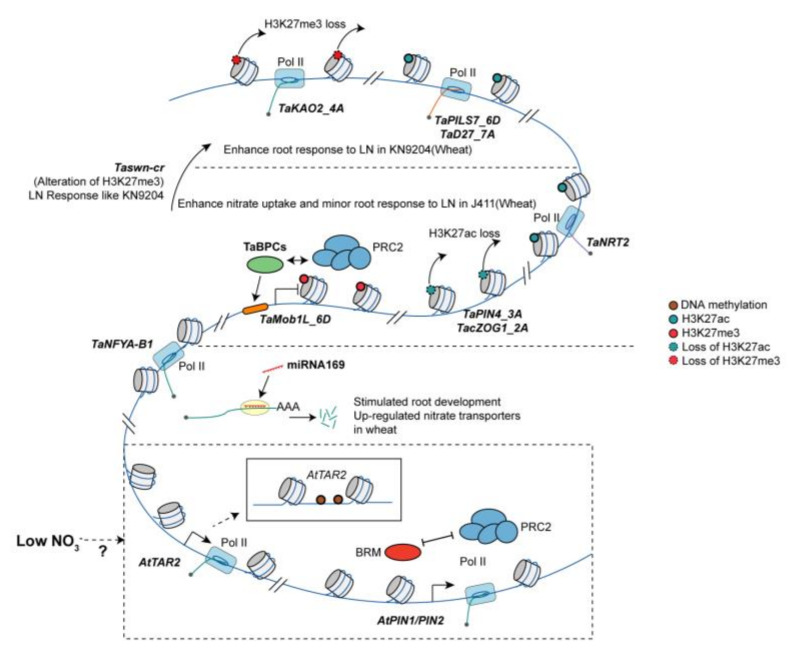
Epigenetic regulation of plant adaptation to LN conditions. Under LN conditions, wheat adopts different adaptation strategies by balancing root system development and nitrate uptake transporters (*NRT2s*). In the KN9204 variety, the loss of H3K27me3 (at *TaKAO2_4A*) and significant gain of H3K27ac enhance the expression of genes related to root development (*TaPILS7_6D* and *TaD27_7A*). Conversely, in J411, there is a greater gain of H3K27me3 and a minor gain of H3K27ac, resulting in reduced root development stimulus but an increase in nitrate uptake transporters (*NRT2s*) through the gain of H3K27ac. Knocking out *TaSWN* (*Taswn*−*cr*) in wheat causes a response to LN similar to KN9204. In addition to histone modifications, miR167 targets *TaNFYA*−*B1*, promoting root development and inducing nitrate transporters in roots under LN conditions. In *Arabidopsis*, *AtTAR2* has also been found to be regulated by DNA methylation, while BRM induces the expression of *PIN1*/*PIN2*, which is repressed by PRC2.

**Figure 3 plants-12-02725-f003:**
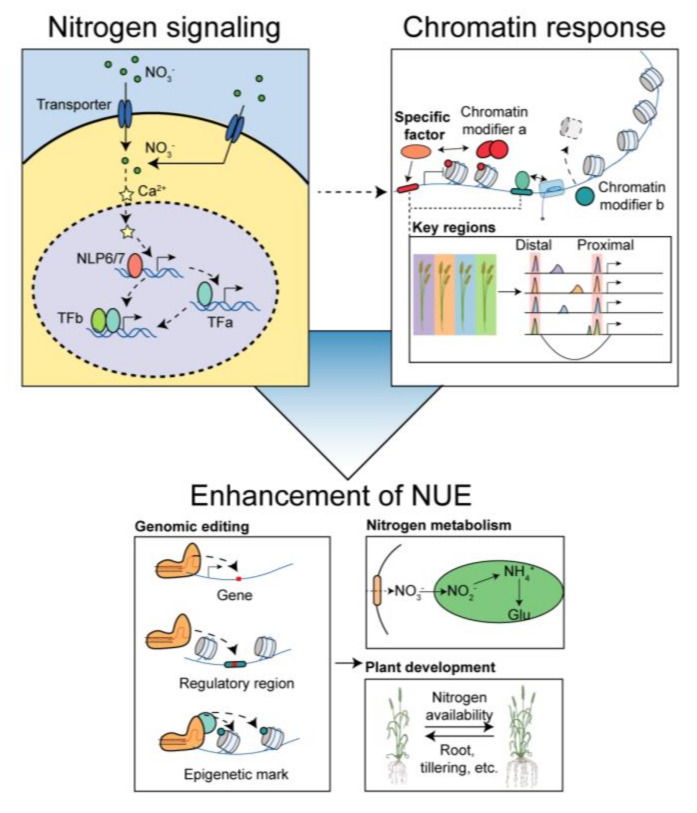
Improving NUE from the epigenetic modification perspectives. The integration of N signaling, chromatin-based regulation, and agronomic improvement holds great promise for enhancing NUE in crop plants. Understanding the dynamic interactions between N−responsive genes, chromatin regulators, and transcription factors at single cell resolution will provide insights into the precise control of gene expression and developmental processes related to N utilization. Integrating population genetics and 3D genomics approaches will enable the identification and characterization of key regulatory regions associated with N response. This knowledge can be leveraged to develop tailored breeding programs for improved agronomic traits with CRISPR tools.

## Data Availability

No new data were created or analyzed in this study. Data sharing is not applicable to this article.

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
