# Peer review of "Epigenetic Regulation of Nitrogen Signaling and Adaptation in Plants"

_plants, 2023, doi:10.3390/plants12142725_

Round 1
Reviewer 1 Report
The article reviews the epigenetic regulation of nitrogen and its role in plant adaptation to low nitrogen environments. Summarizing how DNA methylation, histone modification and microRNAs participate in N response and regulate plant tolerance to low nitrogen. New strategies have been provided to improve nitrogen utilization efficiency and crop productivity in crop cultivation.
However, there are still some issues that need to be improved in the article
First, the regulatory regions and their corresponding gene targets mentioned in conclusion 4.3 do not seem to have been described in detail in the previous text. In addition, paragraph 3.3 mentioned that DNA methylation enables plants to pass on their tolerance to low nitrogen environment to their offspring, and what about other types of epigenetic regulation?
Readable and understandable
Reviewer 2 Report
It is a good effort, but lack structure. You need to cover this by: mono/dicot ( big difference), process (uptake, root mods, signalling, response of vegetative tissues, flowering/seed yield), type of mods, DNA meth, histone mod, ncRNAs, chromatin.
Some specific examples:
“…epigenetic status of the chromatin…” – there is no such thing; chromatin is an “epigenetically controlled DNA folding and expression”, rewrite
“In addition to auxin biosynthesis, auxin transport have been found to processes undergo epigenetic regulation.” – what are you trying to say?
Review needs structure; for example, if you started with signalling, next one would be – changes in roots, then the rest of the plant for example, then may be the effect on flowering. For example, in section 3. Epigenetic regulation facilitates LN adaptation, you first cover the role of chromatin in root architecture, then DNA methylation in root growth, and then you jump to DNA methylation to transgenerational effects. Structure it by covering, DNA methylation, then histone mods, then miRNA/lncRNA (these play an essential role) and then chromatin structure in each developmental process or process of signalling, accumulation etc.
“4. Conclusion and perspective” – why is this named Conclusion, if there is a large section?
“4.4. Understanding the N response at single-cell resolution” – why is this specific methodology in a separate section – this is not a specific type of epigenetic mod or process in the plant
Minor proofreading is required.
Round 2
Reviewer 2 Report
I like this version a lot more. There are still small gaps in the coverage, for example when you talk about root architecture change in response to N, you only cover ncRNAs. Later on, when you talk about LN, you cover methylation/histones. Not sure why.
Somehow, a balance should be found. Because, there can not be any signalling to "normal" level of nitrogen, because that is what plant is adapted to. Plants can only respond to low or high N, and thus it does not make sense to separate sections to N response and LN response.
There are still some issues with English, especially in the new sections.
